# Association of HOMA-IR Versus TyG Index with Diabetes in Individuals Without Underweight or Obesity

**DOI:** 10.3390/healthcare12232458

**Published:** 2024-12-05

**Authors:** Bokun Kim, Gwon-Min Kim, Up Huh, Juhyun Lee, Eunji Kim

**Affiliations:** 1Future Convergence Research Institute, Changwon National University, Changwon 51140, Republic of Korea; fabulousbobo79@changwon.ac.kr; 2Human Community Renovation Research Center, R Professional University of Rehabilitation, Tsuchiura 300-0032, Ibaraki, Japan; rlarnjsals47@gmail.com; 3Medical Research Institute, Pusan National University, Busan 49241, Republic of Korea; 4Department of Thoracic and Cardiovascular Surgery, Pusan National University School of Medicine, Biomedical Research Institute, Pusan National University Hospital, Busan 49241, Republic of Korea; tymfoo82@gmail.com; 5Department of Thoracic and Cardiovascular Surgery, Samsung Medical Center, Seoul 06351, Republic of Korea; juhyun.yi@samsung.com

**Keywords:** diabetes, insulin resistance, metabolically obese but normal weight, obesity

## Abstract

**Background:** There are higher numbers of Asian individuals with metabolically obese, normal weight (MONW) who are susceptible to diabetes than other ethnic groups. The homeostatic model assessment for insulin resistance (HOMA-IR) has been used for years to evaluate IR; however, the triglyceride-glucose (TyG) index has been highlighted as a reliable and cost-effective insulin resistance indicator. Thus, this study explored the association of the HOMA-IR versus the TyG index with diabetes in individuals without underweight or obesity. **Methods:** This cross-sectional study included 10,471 Korean individuals whose body mass index was ≥18.5 and <25.0 kg/m^2^. Depending on metabolic syndrome criteria, subjects with no conditions, one or two conditions, and three or more conditions were assigned to the metabolically healthy and normal weight (MHNW), pre-MONW, and MONW groups, respectively. Diabetes was diagnosed based on HbA1C and medication information. **Results:** The HOMA-IR and TyG index discriminated between each group, showing an increasing trend from the MHNW group to the MONW group. However, this trend was more robust for the TyG index. The cut-off values in the TyG index and HOMA-IR were >8.9 and >1.6 in males and >8.7 and >1.6 in females, respectively. The TyG index’s area under the curve was higher than that of the HOMA-IR in both sexes. MONWs distinguished by the cut-off values of the TyG index and HOMA-IR were 2.370 and 1.726 for males and 2.249 and 1.710 for females times more likely to develop diabetes. **Conclusions:** The TyG index showed a stronger association with diabetes in Korean individuals without underweight or obesity than the HOMA-IR.

## 1. Introduction

Diabetes is a global health concern due to its high incidence rates and resulting complications [1,2]. As reported by the International Diabetes Federation, in 2021, approximately 537 million individuals worldwide were living with diabetes, accounting for about 10.5% of the global population [3]. This condition resulted in healthcare expenditures amounting to USD 966 billion [3]. By 2045, diabetes cases are projected to rise to 783 million, with healthcare costs expected to exceed USD 1054 billion [3]. Thus, significant improvements in screening and detection in the early stages and proper management of diabetes after diagnosis are essential to prevent or suppress the progression of the disease and reduce the number of severe cases.

For decades, obesity has been highlighted as a significant risk factor for diabetes [4,5,6]. Body mass index (BMI) has been broadly adopted to evaluate obesity, and a positive relationship between BMI and diabetes incidence rates is widely recognized [7,8,9]. However, the limitations of BMI have been repeatedly reported; BMI cannot clearly reflect overall fat mass. In particular, it cannot discriminate visceral fat, which induces the onset of insulin resistance (IR) [8,10,11,12]. Recently, it has been noted that IR rather than BMI-based obesity evaluation is strongly linked to several health concerns, such as diabetes and related complications [13]. Based on this consideration, another classification of obesity, “metabolically obese, normal weight (MONW)”, was proposed [14]. Individuals with MONW have increased IR and are highly susceptible to diabetes in spite of their weight being less than 25 kg/m^2^ in terms of BMI [15,16]. It has been recently underscored that there are higher numbers of Asian MONW individuals with diabetes compared to other ethnic groups, and their all-cause mortality is more elevated than obese individuals without IR [17,18,19].

The hyperinsulinemic-euglycemic clamp is regarded as the gold standard to evaluate IR; however, it is a costly, labor-intensive, and time-consuming method [20,21]. The homeostatic model assessment for insulin resistance (HOMA-IR) has been used to evaluate IR in clinical practice since it was first suggested in 1985 [22]. However, the limitation of the HOMA-IR is that insulin, an essential parameter used to calculate HOMA-IR score, is expensive to confirm [21]. In 2008, the triglyceride-glucose (TyG) index was first proposed, and it has been reported to be a valid and reliable index for the assessment of insulin resistance; the TyG index is significantly associated with the hyperinsulinemic-euglycemic clamp [23,24]. It outperforms the HOMA-IR in identifying several health concerns associated with IR [25,26]. In addition, from an economic point of view, the TyG index is more cost-effective for evaluating IR than other methods, as triglycerides and glucose are basic parameters of health check-ups in many countries [16], which means the TyG index is more accessible in developing nations than the HOMA-IR.

Thus, we hypothesized that the TyG index has a deeper relationship with diabetes in individuals without underweight or obesity compared to the HOMA-IR. Based on this hypothesis, we conducted a population-based cross-sectional study to comparatively explore the association of the TyG versus HOMA-IR index with diabetes in individuals without underweight or obesity. The present study aims to provide a more reliable and cost-effective index to evaluate the likelihood of MONW-related diabetes in resource-constrained settings and developing nations.

## 2. Materials and Methods

### 2.1. Study Design and Subjects

For this cross-sectional study, we utilized secondary data from the Korea National Health and Nutritional Examination Survey (KNHANES) from 2015 and between 2019 and 2021, which provides insights into the general health, nutritional status, and lifestyle of the South Korean population. A total of 10,471 subjects (4298 men and 6173 women) aged ≥ 20 years who were not underweight or obese (BMI ≥ 18.5 and ≤ 25.0 kg/m^2^) were included in the analysis (Figure 1). Each subject read and signed an informed consent form approved by the Korea Centers for Disease Control and Prevention (2018-01-03-C-A, 2018-01-03-2C-A, 2018-01-03-5C-A). This study was conducted in accordance with the principles of the Declaration of Helsinki. The current study protocol was approved by the Institutional Review Board of Changwon National University (approval code: 7001066-202404-HR-056, approval date: 19 September 2024).

### 2.2. Assessments

Each subject’s height was assessed to the nearest 0.1 cm in a standing position without shoes, and body mass was determined to the nearest 0.1 kg using digital scales, with subjects dressed in lightweight clothing. BMI was computed as the ratio of weight to height squared (kg/m^2^) [27]. Waist circumference was recorded to the nearest 0.1 cm using a flexible glass fiber tape. Systolic and diastolic blood pressure measurements were taken manually on three separate occasions within a mobile health-monitoring unit. The mean values were calculated and reported in mmHg. Blood samples were obtained in the morning following an overnight fast lasting at least eight hours [16,28]. The levels of glucose, triglycerides (TG), and high-density lipoprotein (HDL) cholesterol in the blood were analyzed using enzymatic and homogeneous enzymatic colorimetric techniques with a Hitachi 7600-210 automatic analyzer (Tokyo, Japan) [16,28]. HbA1C and insulin levels were confirmed using high-performance liquid chromatography and electrochemiluminescence immunoassays, respectively, using the Tosoh G8 (Tosoh, Japan) and Modular E801 (Roche, Germany), respectively [16].

### 2.3. Metabolically Obese but Normal Weight, Triglyceride-Glucose Index, and Diabetes Evaluation

In this study, MONW was defined as a metabolic syndrome (MS), because MS is a collection of metabolic abnormalities, and this approach has been employed for the detection of MONW individuals in several previous studies. MS was defined using the criteria recommended in 2022 by the Korean Society for the Study of Obesity [29], which requires the presence of three or more of the following components:➀Waist circumference (WC) of ≥90 cm in men and ≥85 cm in women➁Systolic blood pressure (SBP) ≥130 mmHg, diastolic blood pressure (DBP) ≥85 mmHg, or medication use➂Fasting plasma triglycerides 150 mg/dL or medication use➃Fasting HDL cholesterol <40 mg/dL in men, <50 mg/dL in women, or medication use➄Fasting plasma glucose ≥100 mg/dL or medication use.

Based on these criteria, subjects with no conditions, one or two conditions, and three or more conditions were assigned to the metabolically healthy and normal weight (MHNW), pre-MONW, and MONW groups, respectively.

The HOMA-IR and TyG index scores were calculated as (fasting plasma insulin × fasting plasma glucose)/22.5 and ln [fasting plasma triglycerides (mg/dL) × fasting plasma glucose (mg/dL)]/2, respectively [22,23]. Diabetes was confirmed when HbA1C ≥6.5 or medication was used.

### 2.4. Statistical Analysis

All data are listed as mean ± standard deviation. An independent *t*-test was used to compare male and female subjects for variables such as HOMA-IR score, HbA1C, body mass index, waist circumference, high-density lipoprotein cholesterol, triglyceride, and insulin, while the Mann–Whitney U-test was applied for age, TyG index, body mass, glucose, and systolic and diastolic blood pressure (Table 1). A one-way ANOVA was used for normally distributed parameters, while the Kruskal–Wallis test was applied for non-normally distributed parameters to compare sex-specific characteristics across the three groups. When significant differences were identified (*p* < 0.05), the Bonferroni post-hoc test was conducted for pairwise comparisons. The Jonckheere–Terpstra test revealed trends in the values among the three groups (two-tailed, *p* < 0.05). The Jonckheere–Terpstra test yielded standardized statistics (SS) that evaluated the strength of the trends in variables that increased or decreased across groups [30,31,32] (Table 2). The sex-specific prevalence of metabolic syndrome and its components across the three groups was compared using the chi-square test, while the linear-by-linear association test was employed to examine trends across the groups (Table 3). The optimal cut-off values for the TyG index to identify MONW in males and females were determined using receiver operating characteristic (ROC) curve analysis. Metrics such as the area under the curve (AUC), sensitivity, and specificity were calculated. These analyses were conducted using MedCalc for Windows Version 9.1.0.1 (MedCalc^®^ Corp, Mariakerke Ostend, Belgium) (Figure 2). A *p*-value of less than 0.05 was considered statistically significant. Logistic regression models were employed to assess the sex-specific associations between MONW and diabetes. The fully adjusted model accounted for potential confounders, including age, educational attainment, household income, smoking status, alcohol consumption, recreational moderate-to-vigorous physical activity, and dietary factors, which are known to influence the relationship with diabetes (Figure 3). Household income was classified into tertiles. Alcohol consumption was classified based on self-reported frequency into four categories: never, up to once per week, 2–3 times per week, or 4 or more times per week. Educational attainment was grouped into three levels: primary education, middle and high school, or college and above. Smoking status was categorized as never smoked, former smoker, or current smoker [33,34]. Recreational moderate-to-vigorous physical activity and nutrition data, including total energy, protein, fat, and carbohydrate intake, were collected using the Global PA Questionnaire and a food frequency questionnaire, respectively (Appendix A). All statistical analyses were performed using the SPSS software Ver. 26.0 (IBM, Inc., Armonk, NY, USA), excluding the ROC curve analysis.

## 3. Results

Table 1 presents the characteristics of the study subjects. The mean age and BMI were 52.5 ± 16.6 years and 22.2 ± 1.7 kg/m^2^, respectively. The mean BMI of males was significantly higher than that of females (*p* < 0.001); however, no difference was found in the mean age between males and females. The mean HOMA-IR and TyG index scores were 1.84 ± 1.83 and 8.58 ± 0.64, respectively, for males and 1.78 ± 1.75 and 8.40 ± 0.58, respectively, for females. No significant differences were found in the HOMA-IR and TyG index scores between males and females. More information on the subjects is provided in the Appendix A.

Table 2 lists the sex-specific characteristics and trends of anthropometric and biochemical parameters among the three groups. In both males and females, the trend test revealed a significant increasing trend in HOMA-IR and TyG index scores and HbA1C levels from the MHNW group to the MONW group (*p* < 0.001 for all). The strength of this increasing trend was higher for the TyG index than the HOMA-IR in both sexes. In addition, the post-hoc test ranked the parameters in ascending order as HOMA-IR, TyG index, and HbA1C, and the differences among the three groups were all significant. Appendix A provides the results of analyses of additional parameters and several covariates in males and females.

The sex-specific prevalence of metabolic syndrome and its components in the three groups is shown in Table 3. In both sexes, the prevalence of elevated triglycerides, glucose levels, high blood pressure, or the use of related medications, along with decreased HDL cholesterol or the use of lipid-lowering medications, showed significant differences among the groups (*p* < 0.001 for all). Additionally, a consistent increasing trend was observed from the MHNW group to the pre-MONW and MONW groups (*p* < 0.001 for all).

Figure 2 compares the ROC curves for the HOMA-IR and TyG indices among males and females with MONW. For males, the optimal sensitivity and specificity thresholds were observed at HOMA-IR >1.6 (AUC: 0.650; sensitivity: 58.70%; specificity: 64.75%), and for females, these thresholds were also found at HOMA-IR >1.6 (AUC: 0.651; sensitivity: 54.82%; specificity: 64.70%) (*p* < 0.001 for both). Regarding the TyG index, the optimal cut-off values were >8.9 for males (AUC: 0.723; sensitivity: 58.70%; specificity: 64.75%) and >8.7 for females (AUC: 0.723; sensitivity: 51.42%; specificity: 85.05%) (*p* < 0.001 for both). For both sexes, the AUC was significantly higher for the TyG index than for the HOMA-IR index.

A comparison of sex-specific odds ratios for the association between the HOMA-IR and TyG indices cut-off values for MONW and diabetes is displayed in Figure 3. For males, in the unadjusted model, relative to MHNW and pre-MONW, the HOMA-IR and TyG index cut-off values for MONW had odds ratios of 1.743 (95% CI: 1.499–2.026; *p* < 0.001) and 2.326 (1.995–2.711, *p* < 0.001) for diabetes, respectively. In the fully adjusted model, relative to MHNW and pre-MONW, the HOMA-IR and TyG index cut-off values for MONW had odds ratios of 1.726 (1.480–2.011, *p* < 0.001) and 2.370 (2.021–2.778, *p* < 0.001) for diabetes, respectively. For females, in the unadjusted model, compared with MHNW and pre-MONW, the HOMA-IR and TyG index cut-off values for MONW had odds ratios of 1.731 (1.503–1.994, *p* < 0.001) and 2.282 (1.978–2.632, *p* < 0.001) for diabetes, respectively. In the fully adjusted model, compared with MHNW and pre-MONW, the HOMA-IR and TyG index cut-off values for MONW had odds ratios of 1.710 (1.479–1.978, *p* < 0.001) and 2.249 (1.937–2.610, *p* < 0.001) for diabetes, respectively. In both sexes, the odds ratios were higher for the TyG index than for the HOMA-IR index, regardless of whether it was adjusted.

## 4. Discussion

This population-based cross-sectional study comparatively explored the association between the TyG versus HOMA-IR index and diabetes in individuals without underweight or obesity. The following findings were obtained. Firstly, the HOMA-IR and TyG indices showed an increasing trend from the MHNW group to the MONW group, but the trend’s strength was more robust in the TyG index than in the HOMA-IR, regardless of sex. Secondly, the AUC of the TyG index was higher than that of the HOMA-IR in both sexes. The optimal cut-off values of the TyG index for predicting the likelihood of MONW were >8.9 and 8.7 in males and females, respectively, and those of the HOMA-IR for predicting the likelihood of MONW were >1.6 and 1.6 in males and females, respectively. Thirdly, subjects in the MONW group, discriminated by the HOMA-IR or TyG index cut-off values, were more likely to develop diabetes than those in the MHNW and pre-MONW groups, regardless of sex. These findings indicate that both the HOMA-IR and TyG indices are associated with the likelihood of diabetes in individuals without underweight or obesity, but the TyG index showed a stronger association, supporting our hypothesis.

In 1981, Ruderman et al. first suggested the concept of MONW [14]. Subsequently, the undesirable consequences of MONW have been well-documented for decades [35,36,37]. In particular, Asians whose BMI is less than 25 kg/m^2^ were reported to have a higher risk of diabetes than other ethnic group [17,38]. Previous studies have explained that Asians have relatively higher visceral fat than other ethnic groups at the same BMI and are more likely to become diabetic [15,17,19]. Therefore, it is critical to determine MONW in Asian individuals to screen and detect high-risk individuals with metabolic obesity-related diabetes. However, a standardized definition of MONW has not been universally agreed upon. Previous studies have characterized MONW using various criteria, including the presence of metabolic syndrome, placement in the highest quartile or a specific range regarding HOMA-IR score, elevated visceral or total fat levels, impaired glucose disposal rates, or the presence of multiple cardiovascular risk factors [17,35,36]. Among the conditions employed in previous studies, IR has been recognized as the primary element. Thus, IR indicators, including HOMA-IR and TyG index scores, can be utilized to confirm MONW in Korean individuals and predict the prevalence of metabolic obesity-related diabetes.

In this study, regardless of sex, the HOMA-IR and TyG indices discriminated between each group, and both showed an increasing trend from the MHNW group to the MONW group. However, the trend’s strength was more robust in the TyG index than in the HOMA-IR (SS = 18.55 versus 28.56 and 23.40 versus 33.91 in males and females, respectively, *p* < 0.001). These results indicate that the TyG index is more closely related to MONW than the HOMA-IR index. In addition, the analysis of the ROC curves supported this finding. The AUC of the TyG index (0.723 in both sexes) was higher than that of the HOMA-IR (0.650 and 0.651 in males and females, respectively) in both sexes (*p* < 0.001). Analysis of the ROC curves showed that the optimal cut-off values of the TyG index for predicting MONW were >8.9 (sensitivity: 51.89% and specificity: 84.69%) and 8.7 (51.42% and 85.05%) in males and females, respectively, and those of the HOMA-IR for predicting MONW were >1.6 (58.70% and 64.75%) and 1.6 (54.82% and 67.70%) in males and females, respectively. Using these cut-off values, the sex-specific odds ratios for the relationship between the HOMA-IR or TyG index cut-off values and diabetes were analyzed. Among male subjects, those in the MONW group, identified according to the HOMA-IR or TyG index cut-off values, were 1.726 and 2.370 times more likely to develop diabetes, respectively, relative to the MHNW and pre-MONW groups (*p* < 0.001). In the case of female subjects, those in the MONW group, identified according to the HOMA-IR or TyG index cut-off values, were 1.710 and 2.249 times more likely to develop diabetes, respectively, than the MHNW and pre-MONW groups (*p* < 0.001). These findings revealed that the TyG index has a stronger association with the likelihood of MONW-related diabetes than the HOMA-IR.

To explain the better performance of the TyG index in evaluating IR in non-obese individuals, the detections associated with TG in this study need to be considered. The strength of the increasing trend from the MHNW group to the MONW group was the strongest in relation to TG (SS: 24.87 and 29.79 in males and females, respectively, *p* < 0.001 for both), while glucose (SS: 24.37 and 29.30 in males and females, respectively, *p* < 0.001 for both) and insulin (SS: 13.53 and 17.49 in males and females, respectively, *p* < 0.001 for both) were the second and third strongest, respectively, regardless of sex. These results indicate that increased TG levels are the primary cause of IR. Mechanistically, TG-rich lipoproteins are hydrolyzed into FFAs and glycerol, which accumulate in non-adipose tissues such as the liver and pancreas. This ectopic fat deposition disrupts insulin signaling pathways by increasing oxidative stress and inflammatory cytokine production, impairing glucose homeostasis and contributing to systemic IR [39,40,41]. This process disrupts the delicate glucose and lipid homeostasis, a hallmark of the progression toward diabetes.

Furthermore, Excessive TG accumulation in the liver, termed hepatic steatosis, further exacerbates IR by inhibiting the insulin-mediated suppression of hepatic glucose production, thereby perpetuating hyperglycemia and metabolic dysfunction [42,43]. Similarly, TG deposition in the pancreas reduces beta-cell function and insulin secretion, vital components of glucose regulation [44]. These dual impacts on insulin sensitivity and secretion underscore TG’s pivotal role in IR development. This mechanistic understanding reinforces the utility of the TyG index as a cost-effective and reliable tool for assessing IR in MONW individuals, offering significant implications for early intervention and risk management.

This study possessed notable strengths as well as certain limitations. Adjustments were made for various potential confounding factors, such as demographic characteristics and lifestyle behaviors, which could influence the relationship between MONW and diabetes. However, although demographic and lifestyle factors potentially affecting the association between MONW discerned according to the HOMA-IR or TyG index cut-off values and diabetes were adjusted for, the conclusion drawn regarding the predictability of the likelihood of MONW discerned according to the HOMA-IR or TyG index cut-off values for diabetes cannot be determined via cross-sectional research. Further longitudinal studies are required to validate our findings. In addition, because all subjects were Korean, it remains unclear whether the study results can be extrapolated to other ethnic groups or nations. Adequate investigations should be performed in various ethnic groups to confirm the conclusions of this study.

## 5. Conclusions

Both the HOMA-IR and TyG indices are associated with the likelihood of diabetes in individuals without underweight or obesity, with the TyG index showing a stronger association. This highlights the TyG index as a reliable and cost-effective tool for assessing MONW-related diabetes risk, particularly in resource-constrained settings and developing nations. Its accessibility compared to the HOMA-IR may enhance early detection and facilitate targeted interventions, thereby improving clinical decision making and diabetes management outcomes.

## Figures and Tables

**Figure 1 healthcare-12-02458-f001:**
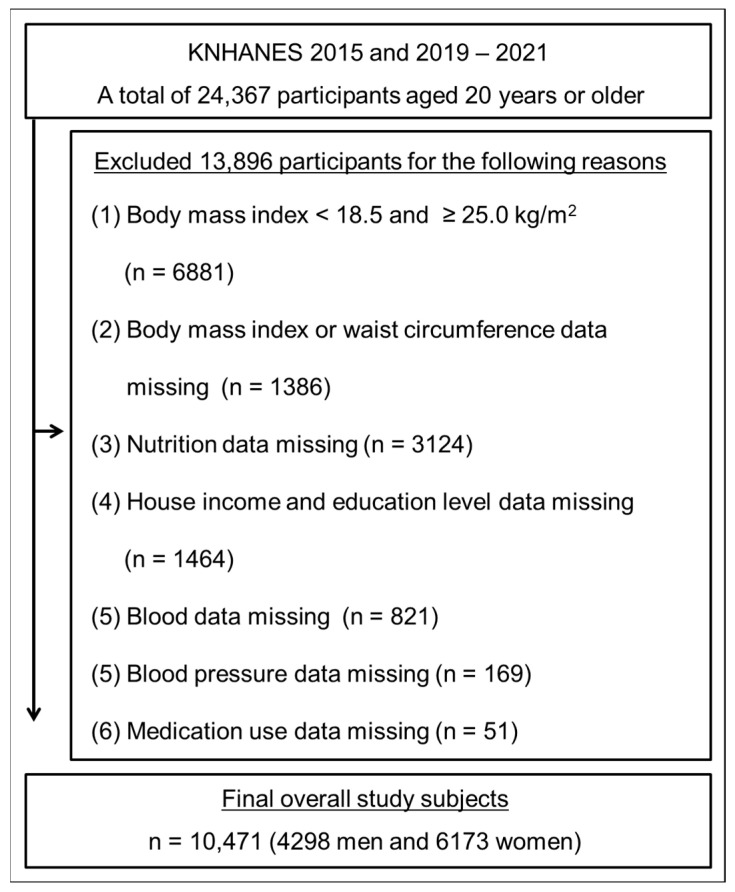
A flowchart of participant recruitment.

**Figure 2 healthcare-12-02458-f002:**
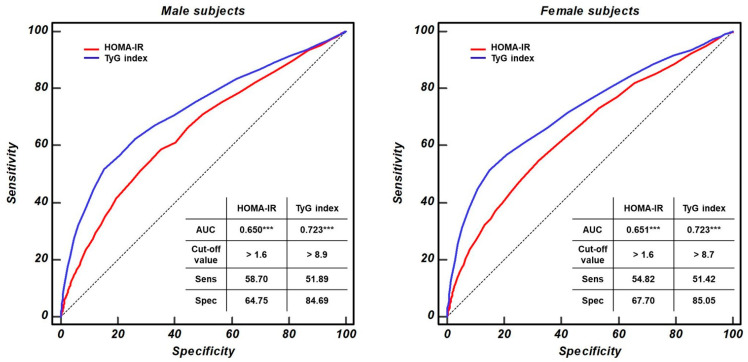
The ROC curves for HOMA-IR and TyG indices in males and females with MONW. *** *p* < 0.001.

**Figure 3 healthcare-12-02458-f003:**
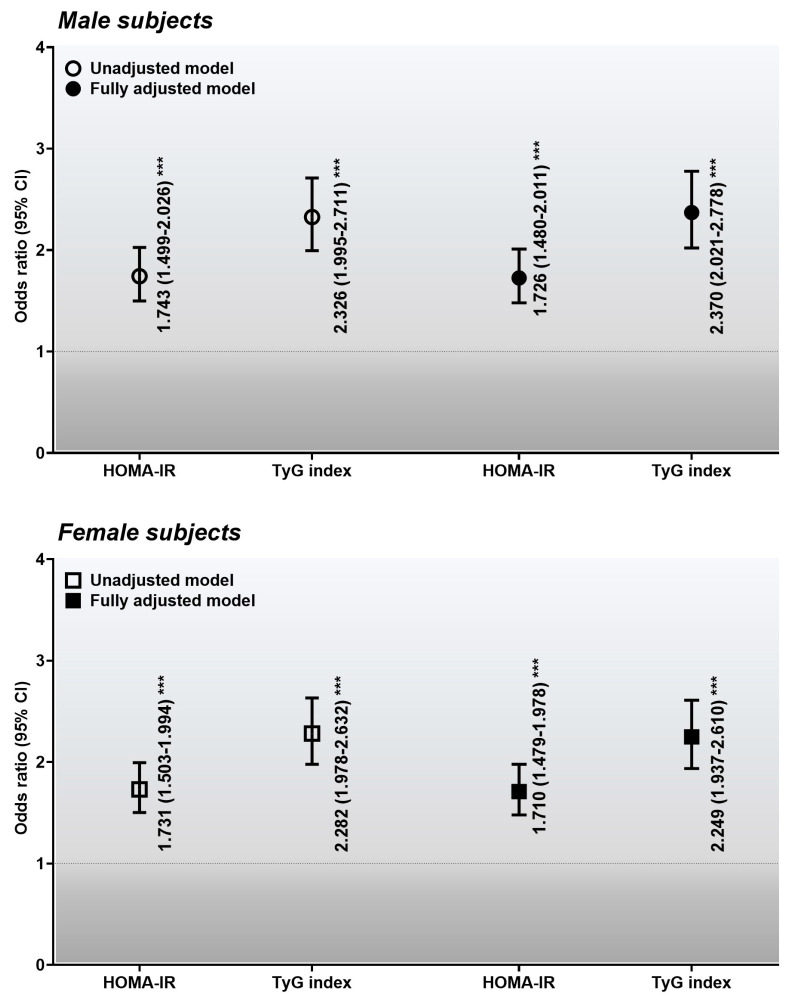
A comparison of sex-specific odds ratios for the association between the HOMA-IR and TyG indices cut-off values for MONW and diabetes. Dotted line: reference value; solid line: 95% confidence interval (CI); uncolored and black circles: ORs for unadjusted and fully adjusted models in males, respectively; uncolored and black rectangles: ORs for unadjusted and fully adjusted models in females, respectively. *** *p* < 0.001 for the ORs for MONW with diabetes, compared with MHNW and pre-MONW. Abbreviations: CI, confidence interval; MHNW, metabolically healthy, normal weight; MONW, metabolically obese, normal weight; OR, odds ratio.

**Table 1 healthcare-12-02458-t001:** Characteristics of study subjects.

	Overall Subjects (n = 10,471)			Male Subjects (n = 4298)			Female Subjects (n = 6173)			*p*-Value for Group Difference
Age, year †	52.5	±	16.6	52.4	±	17.0	52.6	±	16.3	0.544
HOMA-IR	1.81	±	1.78	1.84	±	1.83	1.78	±	1.75	0.095
TyG index †	8.47	±	0.61	8.58	±	0.64	8.40	±	0.58	0.001
HbA1C, %	5.75	±	0.83	5.82	±	0.89	5.70	±	0.78	0.001
Height, cm †	162.7	±	8.9	165.7	±	8.9	160.6	±	8.3	0.001
Body mass, kg †	59.0	±	8.2	61.6	±	8.5	57.1	±	7.4	0.001
Body mass index, kg/m^2^	22.2	±	1.7	22.4	±	1.7	22.1	±	1.7	0.001
Waist circumference, cm	79.2	±	7.1	81.0	±	7.0	78.0	±	6.8	0.001
Glucose, mg/dL †	99.0	±	20.9	101.2	±	22.9	97.5	±	19.2	0.001
High-density lipoprotein cholesterol, mg/dL	53.9	±	13.1	52.2	±	13.1	55.0	±	12.9	0.001
Triglyceride, mg/dL	115.7	±	87.8	105.5	±	1.6	71.6	±	0.9	0.001
Insulin, uIU/mL	7.2	±	5.4	7.1	±	5.2	7.2	±	5.6	0.813
Systolic blood pressure, mm Hg †	119.8	±	16.8	121.1	±	15.9	119.0	±	17.4	0.001
Diastolic blood pressure, mm Hg †	75.1	±	9.8	76.2	±	10.0	74.2	±	9.6	0.001
Medication										
Diabetes, n (%)	1017 (9.7)			478 (11.1)			541 (8.8)			0.001
Diabetes prevalence, n (%)	1788 (17.1)			869 (20.2)			919 (14.9)			0.001

Values are means ± SD. † Mann–Whitney U-test was applied to assess the difference between groups.

**Table 2 healthcare-12-02458-t002:** Sex-specific characteristics and trends of anthropometric and biochemical parameters among the three groups.

	A. MHNW Mean ± SD (95% CI)				B. Pre-MONW Mean ± SD (95% CI)				C. MONW Mean ± SD (95% CI)				*p*-Value for Group Difference	Post-Hoc	SS ^‡^	*p*-Value for Trend ^‡^
Male subjects (n = 4298)	n = 804				n = 2305				n = 1189							
HOMA-IR ^†^	1.29	±	0.70	(1.24, 1.34)	1.79	±	2.04	(1.71, 1.87)	2.32	±	1.82	(2.22, 2.43)	0.001	A < B < C	18.55	0.001
TyG index ^†^	8.15	±	0.39	(8.12, 8.18)	8.53	±	0.55	(8.51, 8.55)	8.95	±	0.72	(8.91, 8.99)	0.001	A < B < C	28.56	0.001
HbA1C, % ^†^	5.48	±	0.53	(5.45, 5.52)	5.79	±	0.83	(5.75, 5.82)	6.10	±	1.08	(6.04, 6.17)	0.001	A < B < C	15.75	0.001
Age, year ^†^	45.6	±	15.5	(44.6, 46.7)	52.2	±	17.2	(51.5, 52.9)	57.3	±	16.0	(56.3, 58.2)	0.001	A < B < C	15.02	0.001
Height, cm	165.3	±	8.7	(164.7, 165.9)	165.6	±	9.0	(165.2, 166.0)	166.1	±	8.8	(165.6, 166.6)	0.137	NS	2.37	0.05
Body mass, kg	59.6	±	8.3	(59.0, 60.2)	61.4	±	8.5	(61.1, 61.8)	63.4	±	8.3	(62.9, 63.9)	0.001	A < B < C	10.09	0.001
BMI, kg/m^2 †^	21.7	±	1.7	(21.6, 21.8)	22.3	±	1.7	(22.2, 22.4)	22.9	±	1.6	(22.8, 23.0)	0.001	A < B < C	15.67	0.001
WC, cm	77.2	±	6.2	(76.8, 77.7)	80.7	±	6.6	(80.4, 81.0)	84.2	±	6.7	(83.8, 84.6)	0.001	A < B < C	22.23	0.001
TG, mg/dL ^†^	81.9	±	28.9	(79.9, 83.9)	119.6	±	89.8	(115.9, 123.2)	176.8	±	141.8	(168.7, 184.8)	0.001	A < B < C	24.87	0.001
HDL-C, mg/dL	58.8	±	11.7	(58.0, 59.6)	53.1	±	12.6	(52.6, 53.6)	45.9	±	12.0	(45.2, 46.6)	0.001	A > B > C	-23.97	0.001
Glucose, mg/dL ^†^	90.4	±	5.7	(90.0, 90.8)	100.3	±	19.9	(99.5, 101.1)	110.4	±	30.7	(108.7, 112.2)	0.001	A < B < C	24.35	0.001
Insulin, uIU/mL ^†^	5.8	±	3.1	(5.5, 6.0)	7.0	±	5.5	(6.8, 7.2)	8.3	±	5.5	(8.0, 8.6)	0.001	A < B < C	13.53	0.001
SBP, mm Hg ^†^	111.3	±	9.6	(110.7, 112.0)	122.4	±	15.9	(121.8, 123.1)	125.0	±	16.6	(124.1, 126.0)	0.001	A < B < C	18.04	0.001
DBP, mm Hg ^†^	72.1	±	6.9	(71.6, 72.6)	77.0	±	10.3	(76.6, 77.4)	77.6	±	10.5	(77.0, 78.2)	0.001	A < B, C	11.06	0.001
Female subjects (n = 6173)	n = 1477				n = 2932				n = 1764							
HOMA-IR ^†^	1.26	±	0.65	(1.22, 1.29)	1.66	±	1.56	(1.61, 1.72)	2.31	±	2.28	(2.21, 2.42)	0.001	A < B < C	23.40	0.001
TyG index ^†^	8.03	±	0.40	(8.01, 8.05)	8.34	±	0.50	(8.33, 8.36)	8.72	±	0.63	(8.69, 8.75)	0.001	A < B < C	33.91	0.001
HbA1C, % ^†^	5.40	±	0.50	(5.37, 5.42)	5.67	±	0.72	(5.65, 5.70)	5.99	±	0.94	(5.94, 6.03)	0.001	A < B < C	22.70	0.001
Age, year ^†^	45.0	±	13.9	(44.3, 45.7)	51.6	±	16.2	(51.0, 52.2)	60.8	±	14.7	(60.1, 61.5)	0.001	A < B < C	28.61	0.001
Height, cm ^†^	161.2	±	7.7	(160.8, 161.6)	160.3	±	8.3	(160.0, 160.6)	160.6	±	8.9	(160.2, 161.0)	0.01	A > B, C	−2.61	0.01
Body mass, kg ^†^	55.8	±	6.7	(55.5, 56.2)	57.0	±	7.4	(56.7, 57.3)	58.3	±	7.9	(57.9, 58.7)	0.001	NS	9.09	0.001
BMI, kg/m^2 †^	21.5	±	1.6	(21.4, 21.5)	22.1	±	1.7	(22.1, 22.2)	22.5	±	1.7	(22.5, 22.6)	0.001	A < B < C	18.05	0.001
WC, cm ^†^	74.3	±	5.5	(74.0, 74.5)	78.0	±	6.4	(77.8, 78.3)	81.2	±	7.1	(80.9, 81.5)	0.001	A < B < C	28.56	0.001
TG, mg/dL ^†^	75.0	±	27.9	(73.6, 76.4)	101.3	±	59.0	(99.1, 103.4)	143.2	±	96.5	(138.7, 147.7)	0.001	A < B < C	29.79	0.001
HDL-C, mg/dL ^†^	62.8	±	10.5	(62.3, 63.3)	54.8	±	12.4	(54.4, 55.3)	48.9	±	12.1	(48.3, 49.5)	0.001	A > B > C	−32.91	0.001
Glucose, mg/dL ^†^	89.3	±	5.8	(89.1, 89.6)	97.0	±	17.8	(96.4, 97.6)	105.2	±	24.9	(104.1, 106.4)	0.001	A < B < C	29.30	0.001
Insulin, uIU/mL ^†^	5.7	±	2.9	(5.6, 5.9)	7.1	±	5.6	(6.9, 7.3)	8.6	±	6.9	(8.2, 8.9)	0.001	A < B < C	17.49	0.001
SBP, mm Hg ^†^	108.5	±	9.8	(108.0, 109.0)	120.0	±	17.8	(119.4, 120.7)	125.9	±	17.7	(125.1, 126.8)	0.001	A < B < C	28.67	0.001
DBP, mm Hg ^†^	70.8	±	7.1	(70.4, 71.1)	75.2	±	10.1	(74.8, 75.5)	75.6	±	9.8	(75.1, 76.1)	0.001	A < B < C	13.63	0.001

Values are means ± SD. † Kruskal–Wallis test was applied to assess the difference among the three groups. ‡ Jonckheere–Terpstra test was used to assess the trends among three groups. NS = not significant; MHNW = metabolically healthy, normal weight; MONW = metabolically obese, normal weight; BMI = body mass index; WC = waist circumference; HDL-C = high-density lipoprotein cholesterol; SBP = systolic blood pressure; SD = standard deviation; DBP = diastolic blood pressure.

**Table 3 healthcare-12-02458-t003:** Sex-specific prevalence of metabolic syndrome and its components in the three groups.

	A. MHNW	B. Pre-MONW	C. MONW	*p*-Value for Group Difference ^#^	*p*-Value for Trend ^##^
Male subjects (n = 4298)	n = 804	n = 2305	n = 1189		
Metabolic syndrome, n (%)	0 (0.0)	0 (0.0)	1189 (100.0)	0.001	0.001
High waist circumference, n (%)	0 (0.0)	283 (12.3)	415 (34.9)	0.001	0.001
High triglyceride or medication, n (%)	0 (0.0)	529 (23.0)	999 (84.0)	0.001	0.001
Low HDL or medication, n (%)	0 (0.0)	398 (17.3)	934 (78.6)	0.001	0.001
High glucose or medication, n (%)	0 (0.0)	962 (41.7)	885 (74.4)	0.001	0.001
High blood pressure or medication, n (%)	0 (0.0)	1137 (49.3)	878 (73.8)	0.001	0.001
Female subjects (n = 6173)	n = 1477	n = 2932	n = 1764		
Metabolic syndrome, n (%)	0 (0.0)	0 (0.0)	1764 (100.0)	0.001	0.001
High waist circumference, n (%)	0 (0.0)	397 (13.5)	626 (35.5)	0.001	0.001
High triglyceride or medication, n (%)	0 (0.0)	488 (16.6)	1448 (82.1)	0.001	0.001
Low HDL or medication, n (%)	0 (0.0)	1049 (35.8)	1602 (90.8)	0.001	0.001
High glucose or medication, n (%)	0 (0.0)	996 (34.0)	1151 (65.2)	0.001	0.001
High blood pressure or medication, n (%)	0 (0.0)	1245 (42.5)	1281 (72.6)	0.001	0.001

Values are subjects’ number (%). ^#^ Chi-square test was employed to assess the difference among the three groups. ^##^ Linear-by-linear association was used to assess the trends among the three groups.

## Data Availability

The datasets analyzed in this study are available from the corresponding author upon request.

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
