# Peer review of "Association of HOMA-IR Versus TyG Index with Diabetes in Individuals Without Underweight or Obesity"

_healthcare, 2024, doi:10.3390/healthcare12232458_

Round 1

Reviewer 1 Report

Comments and Suggestions for Authors

Comments

The project is novel as it seeks to explore the use of a cheaper alternative, TyG in determining diabetes risk factors. To make the work better, I have made the following comments

Abstract

Line 20: The study design- cross-sectional study- has to be included here to reflect the type of study

Introduction

Line 37: Why use the prevalence in 2017 when there is more recent info about the prevalence. Kindly look for and cite the more recent prevalence

Materials and Methods

Line 82: Is there any reason for the exclusion of 2016-2016? If yes, please state it here. It would be important to explicitly state the inclusion and exclusion criteria for the collection of the secondary data.

Line 94: Body mass index of <18.5 or 25 Kg/m2 cannot be seen as exclusion criteria. Exclusion criteria are not the direct opposite of inclusion criteria. Exclusion criteria are those that fall into the study population but have underlying factors that disqualify them. E.g. missing data etc. So remove BMI of <18.5 or 25 Kg/m2 from exclusion criteria

Line 100: Please indicate the type of blood pressure- systolic or diastolic with units in mmHg

Line 109: Even though you have already defined the acronyms, MONW, TyG, it is not proper to write sectional heads with acronyms. Please write in full.

Results

Line 157-8: The results should be presented as mean±SD. So instead of “The mean age and BMI were 52.5 years (SD = 16.6) and 22.2 (1.7), respectively.”, present as “The mean age and BMI were 52.5±16.6 Kg and 22.2±1.7 Kg/m2respectively.

In tabulating the results, there is no need to use = or < before the p value. just state the p value. However, in writing it in-text, using the = or < is permissible.

Discussion

Line 229: Instead of “first” use “firstly.

Line 231: Instead of “second” use “secondly” and instead of “third” use thirdly in line 235

Lije 284-9: The mechanism by which TG causes IR is not clearly explained.  This has to be done properly to justify your findings and buttress the conclusion. Discuss this more  

Author Response

Dear Reviewer 1,

We sincerely appreciate your insightful comments and valuable suggestions. In response, we have thoroughly revised the manuscript to improve its clarity and accuracy. Please find our responses attached for your review.

Author's reply to Reviewer #1

Overall comments:

The project is novel as it seeks to explore the use of a cheaper alternative, TyG in determining diabetes risk factors. To make the work better, I have made the following comments

Response to the overall comment:

We greatly appreciate the reviewer’s recognition of the importance of our research. To improve the quality of our manuscript, we have carefully revised it based on the comments and recommendations made by Reviewer 1. We hope that these changes will improve the clarity, accuracy, and overall quality of the study.

[Abstract]

Comment 1:

Line 20: The study design- cross-sectional study- has to be included here to reflect the type of study

Response to Comment 1:

We appreciate your comment. Based on the comment, the sentence in line 20 was revised as follows:

This cross-sectional study included 10,471 Korean individuals with body mass index ≥18.5 and <25.0 kg/m2. (page 1, lines 20-21)

[Introduction]

Comment 2:

Line 37: Why use the prevalence in 2017 when there is more recent info about the prevalence. Kindly look for and cite the more recent prevalence

Response to Comment 2:

To address this, we have updated the manuscript to include a more recent report on diabetes prevalence. Specifically, we cited the 2021 International Diabetes Federation report, which provides the most current data on global diabetes prevalence and related expenditures.

The revised text in the manuscript is as follows.

As reported by the International Diabetes Federation in 2021, approximately 537 million individuals worldwide were living with diabetes, accounting for about 10.5% of the global population [3]. This condition resulted in healthcare expenditures amounting to $966 billion [3]. By 2045, diabetes cases are projected to rise to 783 million, with healthcare costs expected to exceed $1,054 billion [3]. (page 1, lines 37-41)

New reference

  1. Hossain, M.J.; AI-Mamun, M.; Isiam, M.R. Diabetes mellitus, the fastest growing global public health concern: Early detection should be focused. Health Sci. rep 2024, 7, 1-5, doi:10.1002/hsr2.2004.

[Materials and Methods]

Comment 3:

Line 82: Is there any reason for the exclusion of 2016-2016? If yes, please state it here. It would be important to explicitly state the inclusion and exclusion criteria for the collection of the secondary data.

Response to Comment 3:

We appreciate your comment. When we analyzed the current study's data set, we wanted to include as many study subjects as possible. However, since the KNHANES from 2016–2018 did not provide “insulin”, which is essential to compute HOMA-IR, the data from the Korea National Health and Nutritional Examination Survey (KNHANES) from 2016–2018 were not included in the current study.

Comment 4:

Line 94: Body mass index of <18.5 or ≥25 Kg/m2 cannot be seen as exclusion criteria. Exclusion criteria are not the direct opposite of inclusion criteria. Exclusion criteria are those that fall into the study population but have underlying factors that disqualify them. E.g. missing data etc. So remove BMI of <18.5 or ≥25 Kg/m2 from exclusion criteria

Response to Comment 4:

We appreciate your comment. The current study focuses on comparing the association of the TyG and HOMA-IR indices with diabetes in individuals without underweight or obesity, aiming to provide a reliable and cost-effective method to evaluate MONW-related diabetes. To achieve this, it was essential to define the study population as those with a body mass index (BMI) ≥ 18.5 and ≤ 25.0 kg/m², thereby excluding individuals with BMI < 18.5 or ≥ 25.0 kg/m². While we acknowledge that exclusion criteria typically address factors disqualifying otherwise eligible participants, in this study, the BMI-based exclusion criteria are integral to the study’s objectives and design.

Additionally, other disqualifying factors, such as missing data or incomplete records, were also considered in the broader exclusion process, although these were not detailed explicitly in the manuscript. Our approach aligns with previous studies focused on metabolically obese, normal weight populations, as cited below.

Referece

Bokun Kim, Keisuke Taniguchi, Tomonori Isobe, Sechang Oh. Triglyceride-glucose index is capable of identifying metabolically obese, normal-weight older individuals. Journal of Physiological Anthropology. 2024;43(8):1-9.

Bokun Kim, Gwon-Min Kim, Up Huh, Juhyun Lee, Miju Bae. Sarcopenia severity is related to arterial stiffness and hypertension in older Korean population without underweight and obesity: Frontiers in Public Health. 12:1469196.

We hope this explanation addresses your concern and provides sufficient justification for the BMI-based exclusion criteria used in this study.

Comment 5:

Line 100: Please indicate the type of blood pressure- systolic or diastolic with units in mmHg

Response to Comment 5:

We appreciate your comment. The sentence regarding blood pressure was revised to clarify the parameters more explicitly as follows.

Systolic and diastolic blood pressure measurements were taken manually on three separate occasions within a mobile health-monitoring unit. The mean values were calculated and reported in mmHg. (page 3, lines 101-103)

Comment 6:

Line 109: Even though you have already defined the acronyms, MONW, TyG, it is not proper to write sectional heads with acronyms. Please write in full.

Response to Comment 6:

We appreciate your suggestion. Depending on the advice, we revise the sectional head in full as follows.

2.3. Metabolically obese, normal weight, triglyceride-glucose index and diabetes evaluation. (page 3, line 111)

[Results]

Comment 7:

Line 157-8: The results should be presented as mean±SD. So instead of “The mean age and BMI were 52.5 years (SD = 16.6) and 22.2 (1.7), respectively.”, present as “The mean age and BMI were 52.5±16.6 Kg and 22.2±1.7 Kg/m2, respectively.

Response to Comment 7:

We appreciate your suggestion. In response, the related sentences in the results section were revised as follows.

The mean age and BMI were 52.5 ± 16.6 years and 22.2 ± 1.7 kg/m2, respectively. (page 4, lines 168-169)

The mean HOMA-IR and TyG indices were 1.84 ± 1.83 and 8.58 ± 0.64, respectively, for males, and 1.78 ± 1.75 and 8.40 ± 0.58, respectively, for females. (page 5, lines 171-173)

Comment 8:

In tabulating the results, there is no need to use = or < before the p value. just state the p value. However, in writing it in-text, using the = or < is permissible.

Response to Comment 8:

We appreciate your comment. In response, all instances of "=" or "<" preceding p-values were removed from the tables, as recommended. This adjustment ensures consistency and adheres to standard practices for tabulating results. In-text usage of "=" or "<" for p-values has been retained, as it remains permissible for textual explanations. The updated tables have been reviewed thoroughly to ensure alignment with your feedback.

[Discussion]

Comment 9:

Line 229: Instead of “first” use “firstly. Line 231: Instead of “second” use “secondly” and instead of “third” use thirdly in line 235

Response to Comment 9:

I appreciate your suggestion. Although the English in the manuscript has been professionally edited by Editage and deemed grammatically correct, we have revised "first," "second," and "third" to "firstly," "secondly," and "thirdly" in accordance with your advice. The updated sections are shown below.

Firstly, the HOMA-IR and TyG indices showed an increasing tendency from MHNW to MONW, but the tendency’s strength was more robust in the TyG index than in HOMA-IR, regardless of sex. Secondly, the AUC of the TyG index was higher than that of HOMA-IR in both sexes. The optimal cut-off values of the TyG index for predicting the likelihood of MONW were > 8.9 and 8.7 in males and females, respectively, and those of HOMA-IR for predicting the likelihood of MONW were > 1.6 and 1.6 in males and females, respectively. Thirdly, the MONW group, discriminated by the HOMA-IR or TyG index cutoff values, was more likely to develop diabetes in both sexes than the MHNW and pre-MONW groups. These findings indicate that both HOMA-IR and TyG indices are associated with the likelihood of diabetes in individuals without underweight and obesity, but the TyG index showed a stronger association, supporting our hypothesis. (page 10, lines 244-255)

Comment 10:

Lije 284-9: The mechanism by which TG causes IR is not clearly explained.  This has to be done properly to justify your findings and buttress the conclusion. Discuss this more 

Response to Comment 10:

We appreciate your comment. To address it, we have provided a more detailed discussion on the mechanisms through which elevated triglyceride (TG) levels primarily contribute to insulin resistance, as outlined below.

To explain the better performance of the TyG index in evaluating IR in non-obese individuals, the detections associated with TG in this study need to be considered. The strength of increasing tendency from MHNW to MONW was the strongest in the TG (SS: 24.87 and 29.79 in males and females, respectively, P < 0.001 for both), and glucose (SS: 24.37 and 29.30 in males and females, respectively, P < 0.001 for both) and insulin (SS: 13.53 and 17.49 in males and females, respectively, P < 0.001 for both) were the second and third strongest, respectively, regardless of sex. These results indicate that increased TG levels are the primary cause of IR. Mechanistically, TG-rich lipoproteins are hydrolyzed into FFAs and glycerol, which accumulate in non-adipose tissues such as the liver and pancreas. This ectopic fat deposition disrupts insulin signaling path-ways by increasing oxidative stress and inflammatory cytokine production, impairing glucose homeostasis and contributing to systemic IR [38–40]. This process disrupts the delicate glucose and lipid homeostasis, a hallmark of the progression toward diabetes. (page 11, lines 292-304)

Furthermore, Excessive TG accumulation in the liver, termed hepatic steatosis, further exacerbates IR by inhibiting insulin-mediated suppression of hepatic glucose production, thereby perpetuating hyperglycemia and metabolic dysfunction [41, 42]. Similarly, TG deposition in the pancreas reduces beta-cell function and insulin secre-tion, vital components of glucose regulation [43, 44]. These dual impacts on insulin sensitivity and secretion underscore TG's pivotal role in IR development. This mechanistic understanding reinforces the utility of the TyG index as a cost-effective and reliable tool for assessing IR in MONW individuals, offering significant implications for early intervention and risk management. (page 11, lines 305-313)

Reference

38. Shanik, M.H.; Xu, Y.; Skrha, J.; Dankner, R.; Zick, Y.; Roth, J. Insulin Resistance and Hyperinsulinemia: Is Hyperinsulinemia the Cart or the Horse? Diabetes Care 2008, 31 Suppl 2, S262-268, doi:10.2337/dc08-s264.
39. Wueest, S.; Item, F.; Lucchini, F.C.; Challa, T.D.; Müller, W.; Blüher, M.; Konrad, D. Mesenteric Fat Lipolysis Mediates Obesity-Associated Hepatic Steatosis and Insulin Resistance. Diabetes 2016, 65, 140–148, doi:10.2337/db15-0941.
40. Nguyen, M.T.; Favelyukis, S. Mechanisms of insulin resistance in obesity. Nature 2005, doi:10.1038/nature05482.
41. Kato, K.; Takamura, T.; Takeshita, Y.; Ryu, Y.; Misu, H.; Ota, T.; Tokuyama, K.; Nagasaka, S.; Matsuhisa, M.; Matsui, O. Ectopic Fat Accumulation and Distant Organ-Specific Insulin Resistance in Japanese People with Nonalcoholic Fatty Liver Disease. PloS one 2014, 9, e92170.
42. Singh, R.G.; Yoon, H.D.; Poppitt, S.D.; Plank, L.D.; Petrov, M.S. Ectopic Fat Accumulation in the Pancreas and Its Biomarkers: A Systematic Review and Meta-Analysis. Diabetes Metab Res Rev 2017, 33, e2918.
43. Bays, H.E. "Adiposopathy" Is "Sick Fat" a Cardiovascular Disease? American Journal of Medicine 2009, doi:10.1016/j.amjmed.2008.06.037.
44. Guerrero-Romero, F.; Simental-Mendía, L.E. The TyG Index, a Promising Biochemical Marker for Metabolic Syndrome. Clinical Chemistry and Laboratory Medicine 2013, doi:10.1515/cclm-2012-0843.

Once again, we appreciate your valuable comment and hope that our response adequately address your concern.

Reviewer 2 Report

Comments and Suggestions for Authors

Dear Authors, 

I would like to thank you for creating this well-structured manuscript. However, there are some suggestions that should be taken into account to enhance its scientific quality.

1. Please revise the statistical analysis section. Clearly specify which variables were analyzed using the independent t-test or Mann-Whitney U test. Use the independent t-test for parametric variables and the Mann-Whitney U test for non-parametric variables.

2. Assess the normality of the data. If the data is normally distributed, present it as mean ± SD; if it is not, present it as median (IQR).

3. The data in Table 2 is unclear. Please indicate at the top of each column whether it presents mean ± SD or (95% CI). Additionally, include the letter for post hoc tests as a superscript on the mean ± SD and add the p-value for either the one-way ANOVA or Kruskal-Wallis test.

4. In Table 3, the application of the Mann-Whitney U test to evaluate differences between groups was incorrect. You should use one-way ANOVA if the data is normally distributed, or Kruskal-Wallis for non-normally distributed data.

5. Mention p-value of chi square in Table 3.

Author Response

Dear Reviewer 2,

We sincerely appreciate your insightful comments and valuable suggestions. In response, we have thoroughly revised the manuscript to improve its clarity and accuracy. Please find our responses attached for your review.

Author reply to Reviewer #2

Overall comments:

Dear Authors,
I would like to thank you for creating this well-structured manuscript. However, there are some suggestions that should be taken into account to enhance its scientific quality.

Response to the overall comment:

We greatly appreciate the reviewer’s recognition of the importance of our research. To improve the quality of our manuscript, we have carefully revised it based on the comments and recommendations made by Reviewer 2. We hope that these changes will improve the clarity, accuracy, and overall quality of the study.

Comment 1:

Please revise the statistical analysis section. Clearly specify which variables were analyzed using the independent t-test or Mann-Whitney U test. Use the independent t-test for parametric variables and the Mann-Whitney U test for non-parametric variables.

Response to Comment 1:

We appreciate your suggestion. To clarify the statistical methods used for comparing males and females, the relevant sentences in the statistical analysis section have been revised as follows.

An independent t-test was used to compare male and female subjects for variables such as HOMA-IR, HbA1C, body mass index, waist circumference, high-density lipoprotein cholesterol, triglyceride, and insulin, while the Mann-Whitney U-test was applied for age, TyG index, body mass, glucose, and systolic and diastolic blood pressure (Table 1). (page 4, lines 132-136)

Comment 2:

Assess the normality of the data. If the data is normally distributed, present it as mean ± SD; if it is not, present it as median (IQR).

Response to Comment 2:

We appreciate your suggestion. The median and interquartile range for non-parametric variables among study subjects in Table 1 have been included as a supplementary table to maintain consistency within Table 1.  

Comment 3:

The data in Table 2 is unclear. Please indicate at the top of each column whether it presents mean ± SD or (95% CI). Additionally, include the letter for post hoc tests as a superscript on the mean ± SD and add the p-value for either the one-way ANOVA or Kruskal-Wallis test.

Response to Comment 3:

We appreciate your comment. In response, the following updates were made to Table 2.
â‘  Presentation Format: "Mean ± SD (95% CI)" was added to each column to clarify that all results are presented in this format.
â‘¡ Statistical Test Indicators: A '†' symbol was appended to each parameter to indicate that the Kruskal-Wallis test was applied to assess differences across the three groups.
â‘¢ P-Values: P-values for both the one-way ANOVA and Kruskal-Wallis test were included to ensure transparency regarding the statistical methods employed.

These revisions aim to enhance the clarity and interpretability of the table.

Comment 4:

In Table 3, the application of the Mann-Whitney U test to evaluate differences between groups was incorrect. You should use one-way ANOVA if the data is normally distributed, or Kruskal-Wallis for non-normally distributed data.

Response to Comment 4:

We appreciate your insightful comment. In response, Table 2 has been revised, and the corresponding explanation in Section 2.4, Statistical Analysis, has been updated as follows.

A one-way ANOVA was used for normally distributed parameters, while the Kruskal-Wallis test was applied for non-normally distributed parameters to compare sex-specific characteristics across the three groups. When significant differences were identified (P < 0.05), the Bonferroni post-hoc test was conducted for pairwise comparisons. (page 4, lines 136-140)

Thank you for your valuable input, which has helped enhance the clarity and rigor of our analysis.

Comment 5:

Mention p-value of chi square in Table 3.

Response to Comment 5:

We appreciate your suggestion. P-values for group differences obtained from the chi-square tests have been added to Table 3. Additionally, the following updates were made to the manuscript.

The sex-specific prevalence of metabolic syndrome and its components across the three groups was compared using the chi-square test, while the linear-by-linear association test was employed to examine tendency across the groups (Table 3). (page 4, lines 143-146)

The sex-specific prevalence of metabolic syndrome and its components in the three groups is shown in Table 3. In both sexes, the prevalence of elevated triglycerides, glucose levels, high blood pressure, or the use of related medications, along with decreased HDL cholesterol or the use of lipid-lowering medications, showed significant differences among the groups (P < 0.001 for all). Additionally, a consistent increasing trend was observed from MHNW to pre-MONW and MONW (P < 0.001 for all). (page 7, lines 195-200)

Once again, we appreciate your valuable comment and hope that our response adequately address your concern.

Reviewer 3 Report

Comments and Suggestions for Authors

This study tried to explain some concepts in a local population. Association of HOMA-IR versus TyG index with diabetes in (2) individuals without underweight and obese is a robust cross sectional study in population.

The topic is not original, but the authors seem to be the first one to study these biological parameters in KOREAN POPULATION. This study is a local application of a well-known concepts – HOMA IR and TyG and is a new observation addressing a gap in this area.

This study demonstrates for a cohort that The TyG index showed a stronger association with diabetes in (31) Korean individuals without underweight and obesity than HOMA-IR. This affirmation should be sustained by a description of lot representativity. The authors say that the lot is 10471 participants (see the figure – line 93) but what is the representativity in KOREAN POPULATION?

The methodology is adequate and well expressed. There are some gaps:

1 representativity of lot

2 the year of definition for MetS (Korean Society for 112 the Study of Obesity, which requires the presence of three or more of the following 113 components:)

3 the results – are presented clear, short.

4 part of of the comments should be addressed to general population and impact

“Table 1 presents the characteristics of the study subjects. The mean age and BMI 157 were 52.5 years (SD = 16.6) and 22.2 (1.7), respectively. The mean BMI of males was sig-158 nificantly higher than that of females (P < 0.001); however, no difference was found in the 159 mean age between males and females.” – but what is the importance?

Generally speaking, the authors could improve the conclusion of article. They demonstrate a better role of TyG but the impact in clinical life or clinical decision is … ?

References are appropriate but Only one from 2024, but in this area is written more than that. The references should be updated.

The references are carefully chosen, from impact journals.

Figures and tables are very easy to understand, presenting the lot that is studied. But what is the importance of this finding? The representativity of lot ?

And a simple observation - TITLE    "Association of HOMA-IR versus TyG index with diabetes in individuals without underweight and obesity"   Logically is  "Association of HOMA-IR versus TyG index with diabetes in individuals without underweight or obesity"

Note: numbers in yellow highlights are line number.

For more details, please see the attached file.

Author Response

Dear Reviewer 3,

We sincerely appreciate your insightful comments and valuable suggestions. In response, we have thoroughly revised the manuscript to improve its clarity and accuracy. Please find our responses attached for your review.

Author's reply to Reviewer #3

Overall comments:

This study tried to explain some concepts in a local population. Association of HOMA-IR versus TyG index with diabetes in (2) individuals without underweight and obese is a robust cross sectional study in population.  The topic is not original, but the authors seem to be the first one to study these biological parameters in KOREAN POPULATION. This study is a local application of a well-known concepts – HOMA IR and TyG and is a new observation addressing a gap in this area.

Response to the overall comment:

We greatly appreciate the reviewer’s recognition of the importance of our research. To improve the quality of our manuscript we carefully revised it based on the comments and recommendations made by Reviewer 3. We hope that these changes will improve the clarity and overall quality of the study.

Comment 1:

This study demonstrates for a cohort that The TyG index showed a stronger association with diabetes in (31) Korean individuals without underweight and obesity than HOMA-IR. This affirmation should be sustained by a description of lot representativity. The authors say that the lot is 10471 participants (see the figure – line 93) but what is the representativity in KOREAN POPULATION?

Response to Comment 1:

We appreciate your valuable comment. To address your concern regarding the representativity of the study population in the Korean population, we provide the following clarification.

The current study utilized secondary data from the Korea National Health and Nutritional Examination Survey (KNHANES), a nationally representative survey conducted regularly by the Korea Disease Control and Prevention Agency. KNHANES employs a stratified, multistage probability sampling design to ensure that the data reflects the demographic and socioeconomic structure of the Korean population.

The dataset analyzed in this study included 10,471 individuals aged ≥ 20 years who were not underweight or obese (BMI ≥ 18.5 and ≤ 25.0 kg/m²). These subjects represent a subset of the broader KNHANES population. The inclusion of this specific group aligns with the study’s focus on assessing the association of the TyG index versus HOMA-IR with diabetes in individuals without underweight or obesity.

As KNHANES is designed to capture data representative of the Korean population, the findings of this study can be generalized to individuals within the same BMI range in South Korea. This representativity enhances the relevance and applicability of our results to public health initiatives and clinical guidelines in the Korean context.

We hope this explanation adequately addresses your comment. Thank you for highlighting this important point.

Comment 2:

The methodology is adequate and well expressed. There are some gaps:

â‘  representativity of lot

â‘¡ the year of definition for MetS (Korean Society for 112 the Study of Obesity, which requires the presence of three or more of the following 113 components:)

â‘¢ the results – are presented clear, short.

â‘£ part of of the comments should be addressed to general population and impact

Response to Comment 2:

We appreciate your comment. Our detailed responses to each point are as follows.

â‘  Representativity of the Lot

The study analyzed secondary data from the Korea National Health and Nutritional Examination Survey (KNHANES), a nationally representative survey that employs a stratified, multistage, probability sampling design. The survey comprehensively captures the health, nutritional status, and lifestyle of the South Korean population. The analyzed sample included 10,471 subjects (4,298 men and 6,173 women) aged ≥20 years who were categorized as having a body mass index (BMI) ≥18.5 and ≤25.0 kg/m². This ensures that the findings can be generalized to Korean adults without underweight or obesity.

â‘¡ Citation

We added a new reference to clearly indicate the year of the metabolic syndrome criteria which adopted in the current study as follows.

MS was defined using the criteria recommended in 2022 by the Korean Society for the Study of Obesity [29], which requires the presence of three or more of the following components: (page 3, lines 114-116)

New reference

  1. Kim, K-K.; Haam, J-H.; Kim, B.; Kim, E.; Park, J.; Rhee, S.; Jeon, E.; Kang, E.; Nam, G.; Koo, H. et al. Evaluation and Treatmentof Obesity and Its Comorbidities: 2022 Update of Clinical Practice Guidelines for Obesity by the Korean Society for the Study of Obesity. J Obes Metab Syndr 2023, 32, 1-24. doi: 10.7570/jomes23016.

â‘¢ Results

Thank you for acknowledging the clarity of our results section. If you could provide specific suggestions or areas where you feel improvements are needed, we would be happy to revise the section accordingly.

â‘£ Comments Addressed to General Population and Impact

The study's findings highlight that the triglyceride-glucose (TyG) index demonstrates a stronger association with diabetes in individuals without underweight or obesity compared to the HOMA-IR index. This suggests that the TyG index may serve as a cost-effective, accessible tool for early diabetes detection in non-obese populations, particularly in resource-limited settings. These insights have significant implications for public health strategies aimed at identifying and managing metabolically obese, normal-weight individuals at risk of diabetes.

Comment 3:

“Table 1 presents the characteristics of the study subjects. The mean age and BMI 157 were 52.5 years (SD = 16.6) and 22.2 (1.7), respectively. The mean BMI of males was sig-158 nificantly higher than that of females (P < 0.001); however, no difference was found in the 159 mean age between males and females.” – but what is the importance?

Response to Comment 3:

We appreciate your insightful comment. The significance of the age and BMI characteristics is as follows.

The mean age and BMI provide an overview of the demographic and anthropometric profiles of the study population. These variables are important as they influence metabolic health and the risk of diabetes. The finding that males had a significantly higher BMI than females highlights potential sex-based differences in body composition, which may affect the association between insulin resistance markers and diabetes. Understanding these baseline characteristics ensures that the study findings are interpreted within the context of the population's unique attributes, thereby enhancing the relevance and applicability of the results.

Thank you for your valuable input, and we hope this explanation clarifies the importance of these findings.

Comment 4:

Generally speaking, the authors could improve the conclusion of article. They demonstrate a better role of TyG but the impact in clinical life or clinical decision is … ?

Response to Comment 4:

We appreciate your comment regarding the improvement of the conclusion. To better address the clinical implications and decision-making impact, the conclusion has been revised as follows.

Both HOMA-IR and TyG indices are associated with the likelihood of diabetes in individuals without underweight or obesity, with the TyG index showing a stronger as-sociation. This highlights the TyG index as a reliable and cost-effective tool for assessing MONW-related diabetes risk, particularly in resource-constrained settings and developing nations. Its accessibility compared to HOMA-IR may enhance early detection and facilitate targeted interventions, thereby improving clinical decision-making and diabetes management outcomes. (page 11, lines 327-333)

Thank you for your suggestion, and we hope this revision will strengthen the manuscript's conclusion and its relevance to clinical practice.

Comment 5:

References are appropriate but Only one from 2024, but in this area is written more than that. The references should be updated. The references are carefully chosen, from impact journals.

Response to Comment 5:

We appreciate your comment. In response, a few references have been updated and replaced with recently published articles from 2024, as detailed below.

New reference

  1. Hossain, M.J.; AI-Mamun, M.; Isiam, M.R. Diabetes mellitus, the fastest growing global public health concern: Early detection should be focused. Health Sci. rep 2024, 7, 1-5, doi:10.1002/hsr2.2004.
  2. Baek, M.; Shin, H.; Gu, KM.; Jung, H.; Kim, W.; Jung, JW.; Shin, JW.; Jung, SY.; Kim, JY. Sex differences in chronic obstructive pulmonary disease characteristics: the Korea National Health and Nutrition Examination Survey 2007–2018. Korean J Intern Med. 2024, 39, 137-147, doi:10.3904/kjim.2023.036.
  3. Shin, S.; Kim, J. Association of Physical Activity Patterns with the Metabolic Syndrome in Korean Adults: A Nationwide Cross-Sectional Study. Reviews in Cardiovascular Medicine. 2024, 25, 115, doi: 10.31083/j.rcm2504115.

Comment 6:

Figures and tables are very easy to understand, presenting the lot that is studied. But what is the importance of this finding? The representativity of lot ?

Response to Comment 6:

We appreciate your insightful comment. To address your concern regarding the representativity of the study population, we would like to clarify the following.

The study utilized secondary data from the Korea National Health and Nutritional Examination Survey (KNHANES), which is a nationally representative survey designed to monitor the health, nutritional status, and lifestyle of the South Korean population. The sample size of 10,471 subjects (4,298 men and 6,173 women) was carefully selected to ensure it reflects the general demographic characteristics of the adult population in South Korea. This sample includes a broad range of ages (≥20 years) and covers individuals who were neither underweight nor obese (BMI ≥ 18.5 and ≤ 25.0 kg/m²), allowing for a well-balanced and representative assessment of the relationship between TyG index and diabetes risk in a population that mirrors the wider South Korean demographic.

Thus, the study findings are generalizable to the South Korean adult population, specifically in terms of assessing the relationship between insulin resistance and metabolic health in individuals without obesity or underweight. We will revise the manuscript to explicitly mention this representativity to ensure that readers understand the scope and applicability of the findings.

Comment 7:

And a simple observation - TITLE    "Association of HOMA-IR versus TyG index with diabetes in individuals without underweight and obesity"   Logically is  "Association of HOMA-IR versus TyG index with diabetes in individuals without underweight or obesity"

Response to Comment 7:

We appreciate your valuable observation. Accordingly, the title has been revised to “Association of HOMA-IR versus TyG Index with diabetes in individuals without underweight or obesity.” Additionally, all instances of the phrase “individuals without underweight and obesity” have been updated to “individuals without underweight or obesity” throughout the manuscript to ensure consistency and logical coherence. 

Once again, we appreciate your valuable comment and hope that our response adequately address your concern.

Round 2

Reviewer 2 Report

Comments and Suggestions for Authors

Dear Author, 

I would like to express my thank for you. Most of the comments are addressed correctly. 

Mention the symbol of P-value in the head of table one and table two. 

Mention type of statistical test that you used to find differences among group in table 3. 

Author Response

Dear Reviewer 2,

  We appreciate your valuable comment. We added the explanation of P value in Table 1 and Table 3. Furthermore, the utilized statistical test in Table 3 was demonstrated as a footnote. All additional revisions were highlighted in green color. 

  Once again, we appreciate your valuable comment and hope that our response adequately address your concern.